# Global branches and local states of the human gut microbiome define associations with environmental and intrinsic factors

Julien Tap [1,6] ✉, Franck Lejzerowicz [2,7], Aurélie Cotillard[1], Matthieu Pichaud[1], Daniel McDonald [3], Se Jin Song[2], Rob Knight[2,3,4,5], Patrick Veiga[1,8,10] & Muriel Derrien [1,9,10] ✉

The gut microbiome is important for human health, yet modulation requires more insight into inter-individual variation. Here, we explored latent structures of the human gut microbiome across the human lifespan, applying partitioning, pseudotime, and ordination approaches to >35,000 samples. Specifically, three major gut microbiome branches were identified, within which multiple partitions were observed in adulthood, with differential abundances of species along branches. Different compositions and metabolic functions characterized the branches' tips, reflecting ecological differences. An unsupervised network analysis from longitudinal data from 745 individuals showed that partitions exhibited connected gut microbiome states rather than over-partitioning. Stability in the *Bacteroides*-enriched branch was associated with specific ratios of *Faecalibacterium:Bacteroides*. We also showed that associations with factors (intrinsic and extrinsic) could be generic, branch- or partition-specific. Our ecological framework for cross-sectional and long-itudinal data allows a better understanding of overall variation in the human gut microbiome and disentangles factors associated with specific configurations.

The human gut microbiome is a complex ecosystem holding promises as a target for human health. Recent studies have shown that environmental and host factors explain less than 20% of the variation in microbial composition[1-4], suggesting significant roles for stochastic factors and ecological rules in gut microbiome assembly (reviewed elsewhere[5-7]). A factor known to influence species distribution and diversity of ecosystems is the history of community assembly[8]. Indeed, members' appearance order in the ecosystem influences the evolution of the ecosystem, a phenomenon known as the priority effe, which has

been studied in animal models[9,10] and in vitro[11] but is underexplored in human due to the lack of datasets that are both large-scale and long-itudinally dense. Altogether, multiple factors contribute to the large inter-individual variability of human gut microbiota across the life span. This large variation among subjects, combined with individua-lized stability, justifies understanding better underlying ecological features to guide microbiome-based preventive or therapeutic approaches. Ordination methods, techniques such as principal coor-dinates analysis that aim at representing or analyzing multivariate data

[1]Danone Nutricia Research, Gif-sur-Yvette, France. [2]Center for Microbiome Innovation, University of California San Diego, La Jolla, CA, USA. [3]Department of Pediatrics, School of Medicine, University of California San Diego, La Jolla, CA, USA. [4]Department of Bioengineering, University of California San Diego, La Jolla, CA, USA. [5]Department of Computer Science and Engineering, University of California San Diego, La Jolla, CA, USA. [6]Present address: Université Paris-Saclay, INRAE, AgroParisTech, Micalis Institute, Jouy-en-Josas, France. [7]Present address: Section for Aquatic Biology and Toxicology, University of Oslo, Oslo, Norway. [8]Present address: Université Paris-Saclay, INRAE, MGP, Jouy-en-Josas, France. [9]Present address: Department of Microbiology, Immunology and Transplantation, Rega Institute for Medical Research, Katholieke Universiteit Leuven, Leuven, Belgium. [10]These authors contributed equally: Patrick Veiga, Muriel Derrien. ✉e-mail: julien.tap@inrae.fr; muriel.derrien@kuleuven.be

in fewer dimensions[12], have been used to visualize and describe microbiome data. In addition, previous studies have attempted to distill down this continuous variation into more meaningful partitions, using either data-driven[13] or a priori approaches[14,15]. Partitioning allows for the capture of more precise associations with both intrinsic factors[16–19] and extrinsic factors, such as diet[1,20,21]. Similarly, partitioning of the gut microbiome has been shown to improve the insight into clinical response to dietary intervention[22] or drug treatment[23]. However, to date, partitions are mostly studied within single cohorts and, therefore, potentially lack a more global landscape observed in the human gut. In addition, gut microbiota may fluctuate differentially between subjects over time, with some individuals showing more stability than others depending on their species composition (e.g. with higher abundance of *Faecalibacterium prausnitzii* and *Bifidobacterium* species)[24]. Indeed, recent and more extensive longitudinal studies showed a variable stability between subjects even in the absence of ecological stressors[24,25], suggesting that some configurations may be more prone to fluctuations. However, to date, the intra- and inter-variation of gut microbiome may be obscured in small and homogeneous cohorts. Here, we performed a large-scale analysis of the human gut microbiome. First, using public metagenomes across lifespan and populations (>17,000 samples from CuratedMetagenomicsData[26], consisting of multiple cohorts but with a wide range of analytical methods), we combined clustering, based on Dirichlet Multinomial Mixture models, and ordination approaches[27]. We identified partitions that could be ordinated along branches, i.e., a continuum of microbiome configurations that connects local ecological states.

Then, we interrogated independent data from the American Gut Project (>16,000 samples)[4], mostly consisting of adults, and mostly cross-sectional, but also utilizing longitudinal data in a subset of subjects. We found that most of the patterns (branches and partitions) identified in CMD adults were replicated in the American Gut Project. A longitudinal analysis of 745 subjects suggested that local partitions are connected by preferred ecological paths corresponding to newly identified gut microbiome branches. Last, we showed that partitions exhibited both common and differential associations with environmental and intrinsic factors. Collectively, we identified branch-specific local states of the human gut microbiome connected by ecological paths, which differ in stability and characteristics for both host and environmental factors.

## Results

### Unsupervised partitioning of the human gut microbiome across lifespan and populations

In the absence of large public data from single cohorts with reduced technical variability, pooled public data are increasingly used to study gut microbiome ecology at various taxonomic levels. We thus exploited the public "curatedMetagenomicData" (CMD, version 3) database[26] of 18,726 fecal metagenomes, including 86 studies across the lifespan, as well as different populations and health conditions (Supplementary Fig. 1a), with inherent large variations in technical parameters. To identify partitions of the human gut microbiome, i.e., possible ecological states, we used the Dirichlet Multinomial Mixtures (DMM) partitioning method since it was extensively used in prior attempts in gut microbiome studies[18,28–33]. In short, DMM models are adapted for compositional data, provide a confidence indicator upon samples' classification, allow for building partitions with different dispersions, and can be used to classify a new sample that was not included in the original modelling.

First, we assessed how the DMM partitioning was sensitive to sample size and the presence of low abundant genera. We tested up to 100 possible partitions ($k = 1$–100) for three computational seeds and two CMD cohorts of ca. 1000 individuals (PREDICT1 and LifeLines-DEEP). We found that a larger sample size led to more partitions using

either the Bayesian Information Criterion (BIC) or the Laplace's index to find the best model fit (Supplementary Fig. 2). In addition, after selecting more-abundant genera (i.e., the top 30 genera), the number of partitions modeled for 1000 randomly selected metagenomes (PREDICT1) increased from 4 to 10 (Supplementary Fig. 2) suggesting that low abundant genera introduced noise in partitioning and reduced the number of partitions. Despite library size variations, 84.7% of samples retained more than 80% of reads when filtered to contain only the most abundant 30 genera. We kept this dataset representing 87.1% of total reads for further analyses to delineate a wider variety of partitions.

Second, we identified 24 partitions using DMM on five CMD subsampled training sets of 3233 metagenomes obtained by combining selections of 30 samples for each sex, age category, and region of birth to balance age categories and alleviate the overrepresentation of North Americans and Europeans in CMD (Supplementary Fig. 1b). For up to 100 possible partitions ($k = 1$–100) and using BIC and Laplace's index as goodness-of-fit measures, a consensus of all five subsampled sets resulted in 24 partitions, referred to as DMM(k24) (Supplementary Fig. 3). The DMM(k24) partitions were more homogeneous (based on theta index, see methods) than DMM(k1) (i.e., no partition scenario), except for partitions m18, m19, and m22 (Supplementary Fig. 4a).

Next, 81% of the samples that were left out during subsampling were then classified into a partition with >80% confidence when using the DMM(k24) model (Supplementary Fig. 4b). Partitions m4, m10, m12, m14, m15, m16, m17, m23, and m24, which showed significant alpha or beta-diversity differences between the training and remaining set, were over-represented in the training set compared to the remaining set (Chi-square test, $p < 0.05$). The whole DMM(k24) set, comprising all samples, remained homogeneous by partitions, as indicated by the few alpha (Shannon index) and beta diversity (Jensen-Shannon Distance) differences with the remaining set of samples not used for DMM modeling (Supplementary Data 1). Hence, we hypothesized that our partitions likely represent local ecological states of the human gut microbiome.

### Human gut microbiome local partitions are ordinated within global branches

We intended to characterize those partitions further. We observed an average variation of up to threefold in the Shannon diversity index across the partitions (Fig. 1a and Supplementary Data 2) but did not find any significant correlations between the top five genera with the highest cumulative weight contribution in the model component and alpha-diversity (Fig. 1b). Here, weight contributions to the DMM model are related to genera relative abundance. Of note, two of the least diverse partitions, m19 and m22, were enriched in infant samples (Fig. 1c and Supplementary Fig. 5). We estimated associations between each genus's respective alpha weight in the DMM model, and Shannon estimated alpha diversity of gut microbiome partitions, using Spearman correlations. Regarding other genera with lower contributions to the DMM model, we found that some genera belonging to the Ruminococcaceae and Lachnospiraceae families were positively associated with partition diversity (Spearman's rank correlation test, $p < 0.05$, Supplementary Data 2). The highest associated genus with partitions diversity was *Ruminococcus* (rho = 0.88, $p < 0.001$). Genera having the highest weights in the DMM model like *Bacteroides*, *Prevotella,* and *Bifidobacterium* were not associated with partitions diversity (Spearman's rank correlation test, $p > 0.05$, Supplementary Data 2). Those results suggest that alpha diversity is not the only factor that could help ordinate these gut microbiome partitions.

Next, we reasoned that those partitions could represent stages of ecological progression, either between or within age categories. We applied the Potential of Heat-diffusion for Affinity-based Trajectory Embedding (PHATE) algorithm to the CMD dataset to further explore this possibility. PHATE is a visualization method conceived to discover latent structures, such as transitions, in high dimensional data while

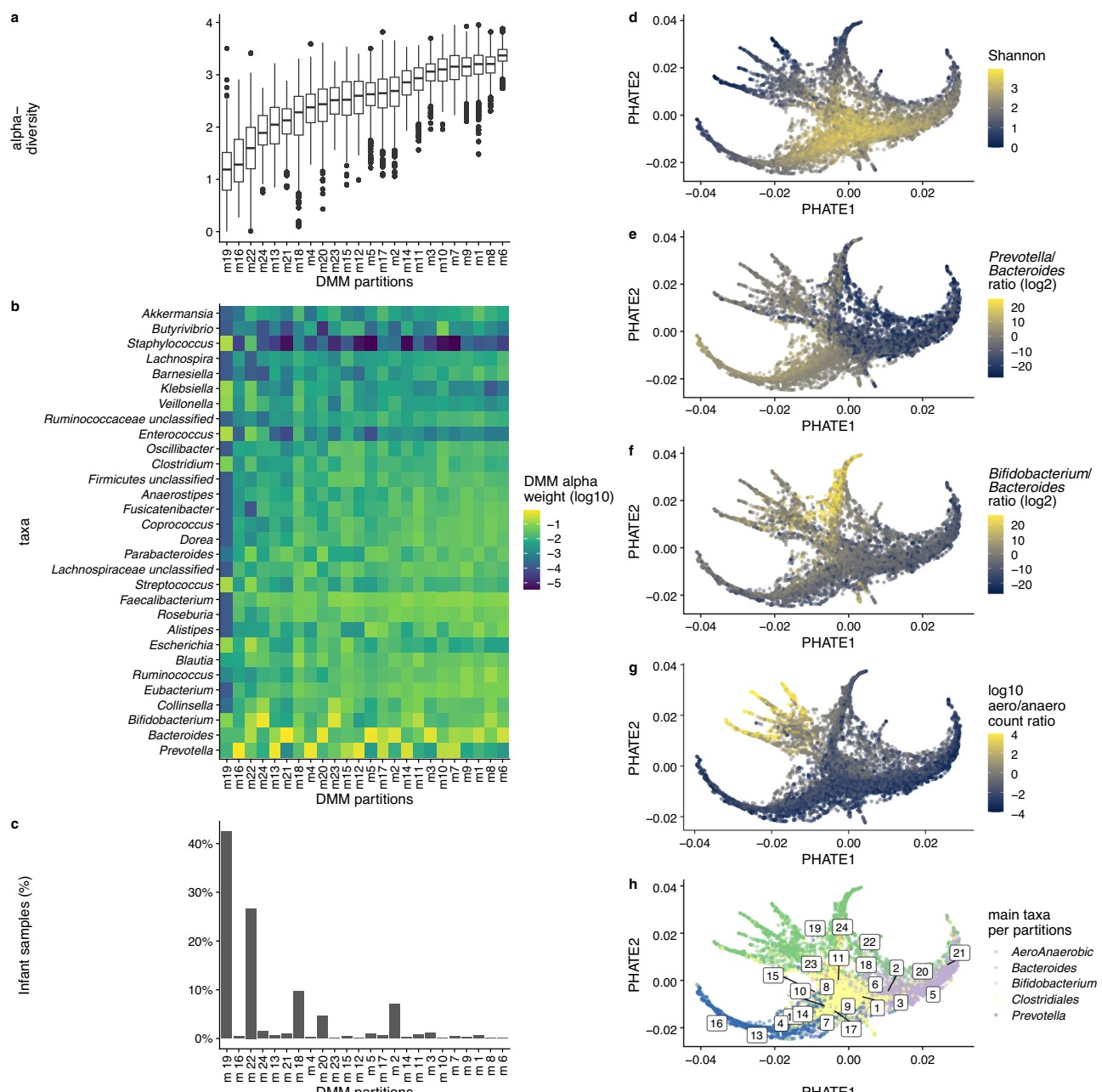

**Fig. 1 | PHATE branches and DMM-based partitions of the human gut microbiome. a** Boxplot for Shannon index in the 24 partitions, ranked by increasing values (*n* = 17,847 samples). The box bounds the interquartile range (IQR) divided by the median, and whiskers extend to a maximum of 1.5 × IQR beyond the box. Dots are sample data points. **b** Heatmap of major genera in the 24 partitions. Partitions are ranked, by increasing values of Shannon index while genera are ranked in increasing cumulative weight across all partitions (i.e., *Prevotella* has the highest cumulative weight). Color accounts for genera alpha weights in the DMM(k24) model. **c** Proportion of infants samples as a function of DMM gut microbiome partitions. **d–g** PHATE scatter plot where samples were colored respectively by Shannon index, *Prevotella:Bacteroides* ratio, *Bifidobacterium:Bacteroides* ratio, and aerobic anaerobic genera count ratio. **h** Mapping of the centroid of the 24 gut microbiome partitions on the branches detected by PHATE. The branches are colored according to main taxa for each partition.

conserving global and local structures of the data, which has previously been applied to gut microbiome data to detect "branches"[27].

The PHATE analysis resulted in several branches (Fig. 1d–h), i.e., continuums of microbiome configurations connecting local ecological states, on which we mapped the centroids of the partitions previously identified. The 24 partitions were distributed along branches indicating congruence between two independent approaches (DMM-based partitioning and PHATE-visualization). Metagenomes and DMM(k24) partitions were sorted similarly following Shannon's alpha diversity gradient along each branch, with the least diverse partitions at the tips of the branches (m19, m16, m24, m21) (Fig. 1d). We also observed that

some branches were associated with ratios of *Prevotella*, *Bacteroides*, and *Bifidobacterium* (Fig. 1e, f), as well as aerotolerant bacteria (represented mainly by Enterobacteriaceae and Streptococcaceae) (Fig. 1g). Individuals older than three years old were less frequent in branches enriched with *Bifidobacterium* and aerotolerant bacteria (Supplementary Fig 5).

Consequently, and in agreement with the cumulative DMM model alpha weight of genera (Fig. 1b), we classified the DMM-based partitions as a function of their most dominant taxa: *Bacteroides*, *Prevotella*, Clostridiales (Lachnospiraceae and Ruminococcaceae genera), *Bifidobacterium*, and aero-anaerobic facultative genera (Fig. 1h). We found

that the projection of the partitions on the PHATE map was consistent with the branches' drivers, such that central partitions were driven by enrichment in genera related to Clostridiales and *Bifidobacterium*, and were among the highest in alpha-diversity (Fig. 1a). A differential analysis at species level (CLR transformed) on adult subjects revealed multiple differences between the central root (M8) and tips of branches (Supplementary Data 3). For instance, the partition closest to the tip in the *Bacteroides* branch (m21) was enriched in *Bacteroides stercoris* and *Clostridium bolteae* while depleted in *Bifidobacterium adolescentis* and *Ruminococcus bromii*. Meanwhile, the partition closest to the tip of *Prevotella* branch (m16) was enriched in *Prevotella coprii*, *Prevotella stercorea* while depleted in *R. bromii* and *Eubacterium hallii*. The partition encompassing the *Bifidobacterium* branch (m24) was enriched in *Bifidobacterium bifidum*, *Bifidobacterium breve* and depleted in *R. bromii* and *Dorea longicatena*. The 'aerobic and aerotolerant' branch was enriched in *Enterococcus faecalis*, *Streptococcus epidermis*, *Streptococcus mitis* and depleted in *R. bromii*, *B. adolescentis*, *F.prausnitzii*. Overall, the core was most often enriched in *B. adolescentis* and Clostridiales (*R. bromii*, *F. prausnitzii*) compared to tips.

We further tested whether branching composition continuums could be retrieved using an alternative and a complementary, pseudo-time method called Wishbone[34], to provide an ordering of samples along the continuous branches. Wishbone is tailored to detect a trajectory based on a t-SNE ordination, that bifurcates from a common root into two branches, and along which the samples are ordered. Hence, we ran Wishbone on a t-SNE ordination computed from 8,356 samples encompassing the main branches (*Bacteroides* and *Prevotella*) as well as the central partition (m8). Consistent with the PHATE analysis, Wishbone detected the central m8 partition as the root from which the two *Bacteroides* and *Prevotella* branches deviated (Supplementary Fig. 6). At genus level, from root to the tips we observed that *Bifidobacterium* and *Ruminococcus* declined along the branches, followed by *Faecalibacterium* while *Bacteroides* and *Prevotella* increased in their respective branch, confirming our observations from the DMM partitions (Supplementary Fig. 7a). Reporting the abundance dynamics at the species level along the Wishbone trajectory revealed a succession for different species of *Bacteroides* (Supplementary Fig. 7b) including *B. ovatus* and *B. fragilis* only appearing at the very end of the Wishbone trajectory. Interestingly, *F. prausnitzii* and *B. bifidum* exhibited a gradual and sharp decrease in abundance along the trajectory, respectively. The partitions were ordered in a similar way to PHATE along these branches, keeping in notably the partitions located respectively on the tips of the two branches. (Supplementary Fig. 7c).

Overall, we showed that DMM-based local partitions can be ordered within the global structure, called branches, revealed by PHATE and confirmed by Wishbone.

## Low-diversity tips of branches display potential functional shifts

To document the functional associations of the extreme ecological states, we identified which metabolic pathways were differentially abundant in the metagenomes composing the low-diversity branch tips with multinomial regressions models (songbird) using the high-diversity partition m8 as a reference (Supplementary Fig. 8). Our analysis indicated that the functional characteristics of the root partition (m8) compared to the m16 (*Prevotella*) and m21 (*Bacteroides*) tip partitions, are largely driven by taxonomic differences, as the root was more strongly associated with functions associated with Archaea such as methanogenesis (or alternatively, a lack of Archaea in the tip partitions). *Prevotella* tip (m16) displayed a higher association with pathways related to nitrate reduction, DHNA (menaquinone) biosynthesis, heme biosynthesis (songbird model $Q^2 = 0.2831$), pathways of the tetrahydrofolate biosynthesis (PWY-7539/PWY-6147), which are known to be involved in the oxidative stress response, overall suggesting that *Prevotella* tip is enriched in pathways related to aerobic and aneror-bic

respirations. The *Bacteroides* branch tip partition (m21) was enriched with amino acid (PWY-5030 and ARGININE-SYN4-PWY) and glycosaminoglycan degradation (PWY-6572) pathways, suggesting an increased capacity for amino acid catabolism in this partition (songbird model $Q^2 = 0.3833$). The m21 partition was also enriched in Gram-negative cell wall biosynthesis (PWY-6478, PWY-6749, PWY-7312, NAGLIPASYN-PWY).

We further used the recently developed Gut Microbiome Health Index (GMHI), which predicts the likelihood of disease based on the distribution of gut microbial species in adults, as a proxy for health[35]. There was generally a higher GMHI along the entire *Prevotella* branch (Supplementary Fig. 9), suggesting that while the tip of the *Prevotella* branch may present both alterations in composition and, therefore, functions of the gut microbiome (reflected by a loss of diversity and a functional shift), the overall likelihood of disease may not necessarily be affected.

## Global and local partitioning can be replicated in the 16S rRNA dataset of the American Gut Project

Given the large amount of available 16S rRNA gene amplicon data with associated metadata in public databases, we reasoned that confirming the validity of our approach on a 16S rRNA dataset would facilitate further investigations aiming to decipher the links between branches or partitions with health and environmental factors. Thus, we applied the same partitioning and ordination approach to the American Gut Project dataset (see methods), which consists of ca. 16,000 16S rRNA gene amplicon sequencing-based fecal samples associated with multiple demographic, lifestyle, health, dietary variables (Supplementary Fig. 1b), and which includes longitudinal sampling from a subset of individuals. Using DMM-based modeling at the genus level (Supplementary Fig. 10a), the best model fit in the AGP database was obtained with 19 partitions. Similar to the CMD partitions, we showed high intra-partitions homogeneity (except for partition 19) (Supplementary Fig. 10b) based on high prediction confidence (Supplementary Fig. 10c) and consistent Shannon alpha diversity between the training and remaining sets (Supplementary Fig. 10d).

We further sought to verify whether AGP partitions were analogous to CMD's. Because the AGP cohort is mainly composed of adult samples (>18 years with prevalence >90%) (Supplementary Fig.1) and is less heterogeneous in terms of analytical parameters, population demographics (age, country of residence), we reasoned that the CMD adult-associated partitions should capture AGP partitions. We compared the partitions between the two datasets using a hierarchical clustering approach (based on common dominant genera (Fig. 2). We did not observe database-specific clustering and instead noticed the pairing of partitions originating from the heterogeneous (i.e., CMD) and single cohort (i.e., AGP), suggesting that analogous partitions could be retrieved regardless of the cohort and technical parameters used (Fig. 2a). Then, we measured the Jensen-Shannon distance between DMM-based partitions alpha weights from both datasets. There was no significant difference between CMD and AGP datasets (PERMANOVA, $p > 0.05$). However, we detected an *Akkermansia*-enriched branch consisting of a single partition (M13) in the AGP dataset (Fig. 2a) that was not detected in the CMD dataset. Then, we generated a PHATE-map of the AGP dataset, which resulted in the generation of three main branches. Similar to the CMD dataset, Shannon's alpha diversity decreased towards the tip of the three branches (Fig. 2b), and a contrasting *Bacteroides to Prevotella* ratio discriminated the major two branches (Fig. 2c and Fig. 1e). The projection of the centroids of the 19 DMM-partitions on this PHATE-map led to a similar result as obtained previously, with partitions being arranged along the global branches (Fig. 2d). It is noteworthy to observe a declining gradient of alpha-diversity from Clostridiales-dominated partitions towards the tips of the *Bacteroides* or *Prevotella* branches: the M1 partition being the most diverse while the least diverse *Bacteroides* and *Prevotella*

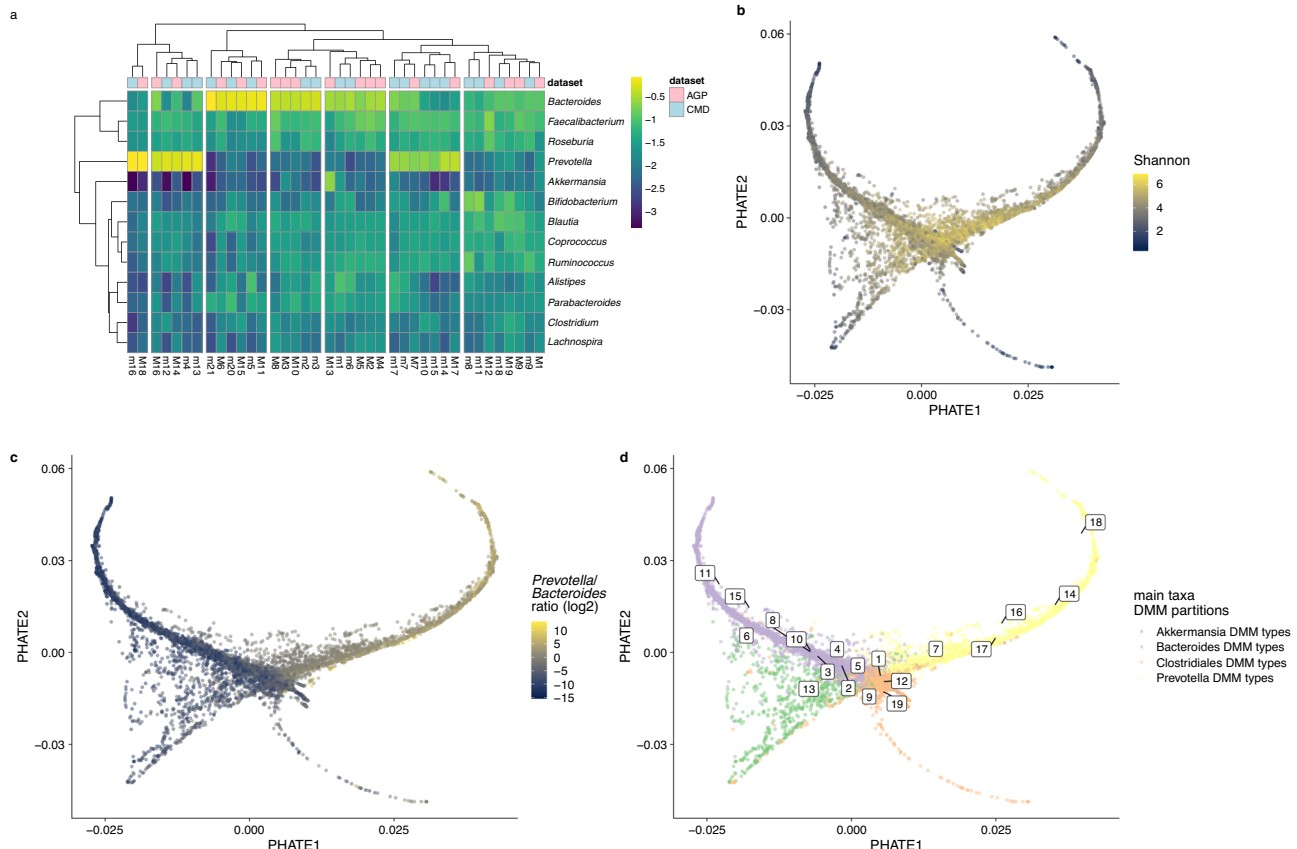

**Fig. 2 | PHATE branches and DMM-based partitions in American Gut project.**
**a** Grouping the partitions according to their similarity in AGP and CMD databases. The colors in the heatmap account for alpha weights in the DMM models (Ward clustering based on Jensen–Shannon distance). **b** The branches are colored according to Shannon diversity and **c** *Prevotella*: *Bacteroides* ratio **d** Mapping of the partitions on the branches. Each partition centroid is mapped on the branches.

partitions, respectively M11 and M18, formed the tips of their branches. Altogether, our analysis of the AGP cohort largely replicated our initial observation indicating that the human gut microbiome configurations are consistent despite analytical variations.

**Local partitions and branches exhibit varying levels of stability**
The ordination of local partitions within global branches prompted us to hypothesize that those partitions represent different states of ecological succession, which we tested with longitudinal data.

We analyzed data originating from 745 participants of the AGP cohort, from whom at least two samples were collected over time with 12.5 (IQR [1.1-73.0]) days apart on average. We assigned each of the associated 2,998 gut microbiome profiles to a partition and a branch and reported the changes between two-time points (Supplementary Fig. 11). Individuals assigned to *Bacteroides* and *Prevotella* branches tended to remain in their initial branches over time while individuals assigned to Clostridiales or *Akkermansia* branches tended to transition to *Bacteroides* and, to a lesser extent, to *Prevotella* branches. (Supplementary Fig. 11).

Then, to test whether local partitions are either random stratification or ecological states, we built a network based on longitudinal data as a function of stability, with instability marked by a high occurrence of observed switches (Fig. 3a). Of note, this network was performed in an unsupervised fashion, (i.e not integrating the partition coordinates depicted in Fig. 2d). The percentage of individuals who remained in their initial partition was, on average 42%, and consistently superior to the frequency computed using randomly generated events (-10% upper CI 95% bound), suggesting that partitions may be relatively stable states (Fig. 3b). Out of the 18 partitions, the M15 and M14 partitions were the most stable within the *Bacteroides* and *Prevotella* branches, respectively.

In contrast, some partitions were connected by a higher occurrence of switches (e.g., between M3 and M5 and between M2 and M6) based on the network. These switches occurred in the *Bacteroides*-enriched branch where *Faecalibacterium* and *Bacteroides* genera had the highest weight in those partitions. These partitions were further assessed to determine whether they resulted from an artifact of over-partitioning. We reasoned that an over-partitioning would result in the same variation over time between and within tested partitions using alpha-diversity and the ratio of the two most abundant genera (*Faecalibacterium* and *Bacteroides*) as markers of the ecosystem composition in the *Bacteroides*-enriched branch. By plotting the variation over time of those parameters, we observed that compositional changes were limited where switching between the partitions did not occur by an individual. Meanwhile, compositional changes were scattered and overlapping where switching did occur between partitions (Supplementary Fig. 12).

We further investigated the dynamics of the *Faecalibacterium:Bacteroides* ratio during M3/M5 and M2/M6 partition switches and observed that the variations of this ratio were elevated in the case of inter-partition switches compared to intra-partition fluctuations (Fig. 3c, d). This observation suggests that the differential ratio of *Faecalibacterium:Bacteroides* is larger between samples of participants that switch between partitions than those that remain stable over time within a partition, supporting our hypothesis that partitions may represent local stable states. Of note, the median values of the *Faecalibacterium:Bacteroides* ratio in subjects that switch partitions over time nearly coincided between the two sets of partitions (M2/M6 and M3/M5), at 0.24 and 0.26, respectively (Fig. 3e, f) possibly suggesting that a *Faecalibacterium:Bacteroides* ratio of 1:4 may be a potential marker of gut microbiome instability (Fig. 3g, h) for the *Bacteroides*-enriched microbiome branch. A ratio below 1:64, corresponding to the

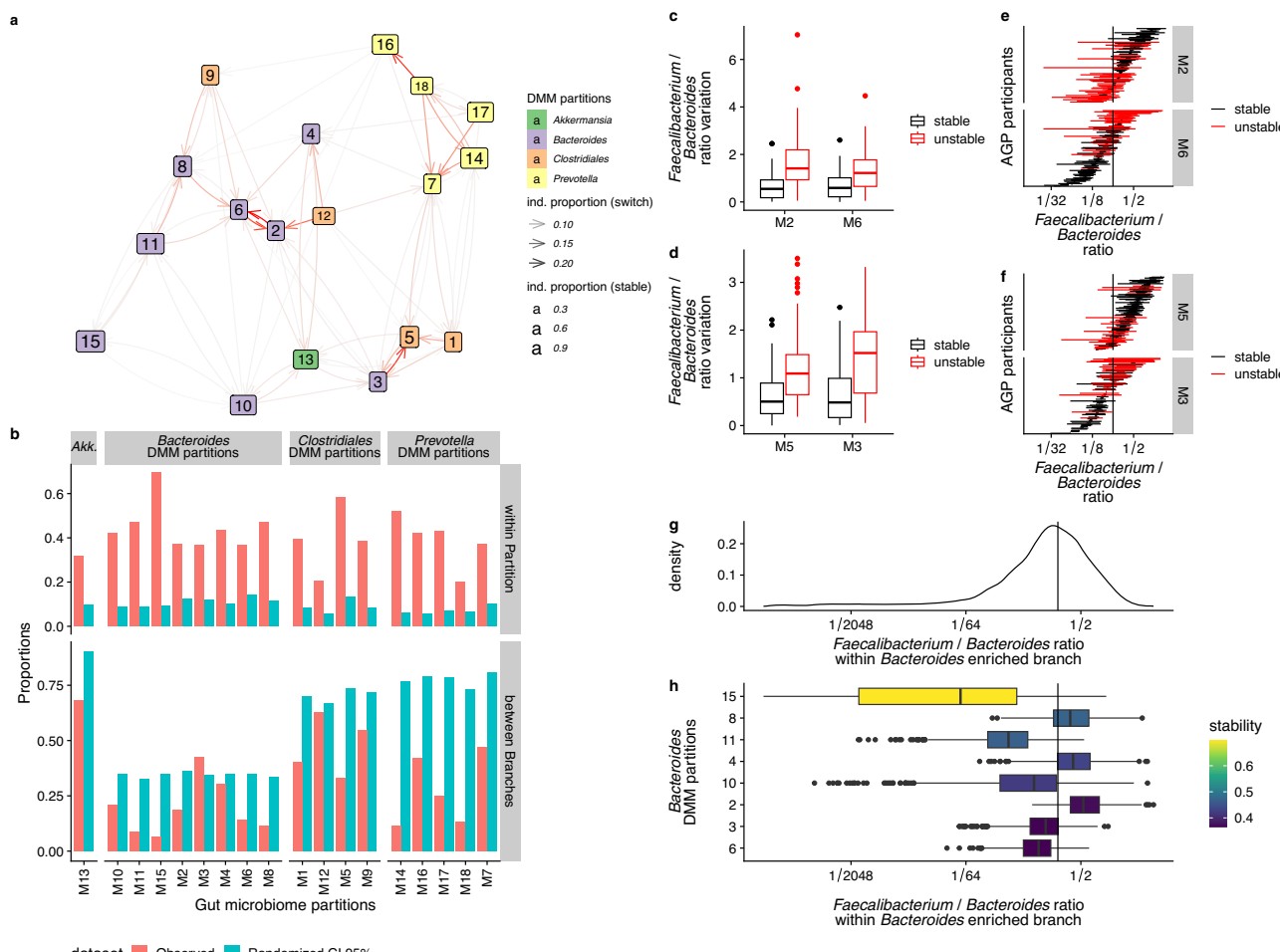

**Fig. 3 | Longitudinal analysis of DMM partitions within the branches. a** A network between partitions shows how individuals can change partition as a function of time. Node size indicates the stability of the partitions, and their colors reflect the corresponding branches. Arrow width represents the proportion of change between each node. Only edges representing more than 5% of the partition events (including switch and non-switch), to decrease visual noise, were shown to represent 81% of the events. **b** The proportion of participants who remain stable within the same partition (top), or switch to a different branch (bottom) for observed and randomly assigned datasets. For the randomized dataset, the worst-case scenario with 95% confidence interval bounds is shown (upper bound for "within partition" and lower bound for "between branch" panels). Boxplot for *Faecalibacterium*:*Bacteroides* ratio variation as a function of the stability between partitions M2/M6 (*n* = 261 samples) (**c**) and M3/M5 (*n* = 250 samples) (**d**). The box bounds the IQR divided by the median, and whiskers extend to a maximum of 1.5 × IQR beyond the

box. Dots are sample data points. *Faecalibacterium*:*Bacteroides* ratio per AGP participants having two successive samples belonging to partitions M2/M6. Red color accounts for subjects in which gut microbiome switched partitions over time. **e** or M3/M5 (**f**) Horizontal black lines accounted for AGP participants that remained stable in the partition over time. Horizontal red lines accounted for AGP participants that switched to another partition over time. The vertical black line shows the average *Faecalibacterium*:*Bacteroides* ratio for AGP participants that switched to another partition over time. **g** *Faecalibacterium*:*Bacteroides* ratio distribution density within Bacteroides-enriched branch. The vertical black line shows the average *Faecalibacterium*:*Bacteroides* ratio for AGP participants that switched to another partition over time. **h** Boxplot for *Faecalibacterium*:*Bacteroides* ratio as a function of Bacteroides-enriched partitions (*n* = 8583 samples). The box bounds the IQR divided by the median, and whiskers extend to a maximum of 1.5 × IQR beyond the box. Dots are sample data points.

median of the m15 partition, the gut microbiome was in an altered and stable state.

## Global and local variations are differently associated with host and environmental factors

Last, we interrogated the AGP dataset (ca 16.000 fecal samples, see methods) to identify associations between branches/partitions and factors relating to the host including dietary habits, lifestyle, region of birth, age, BMI, bowel movement frequency, sex, diseases and antibiotic history). Given the high dimensional nature of the data, we fitted a multinomial logistic regression across 100 predictors (i.e., factors) collected through the AGP main questionnaire (FDR < 0.1) (Supplementary Data 4). To gain insights on factors that could explain the decreased diversity of the communities along the branches, we used the most central and diverse partition (M1) as the reference in this logistic regression, which was primarily composed of female

participants consuming vegetables in high frequency (daily), with low exposure to antibiotics (Fig. 4a). For each predictor, the model returned odds ratios representing the strength of association for a given partition (vs. the reference one).

Overall, host predictors (age, BMI, bowel movement frequency), antibiotic history, and region of birth had the highest weight (i.e., absolute log odds ratio) in the model (Fig. 4b), while dietary predictors contributed to a lesser extent. Among the dietary factors, plant diversity, the frequency of snacks, vegetables, and sugary sweets had the highest weights (Fig. 4b).

Next, we estimated how the weight of these predictors varied across partitions along their respective branches. A higher cumulative weight of predictors was observed for partitions with lower alpha-diversity and closer to the tip within each branch (Fig. 4c). This observation indicated that the more distant the partition was from the reference partition M1 the more it was associated with host and

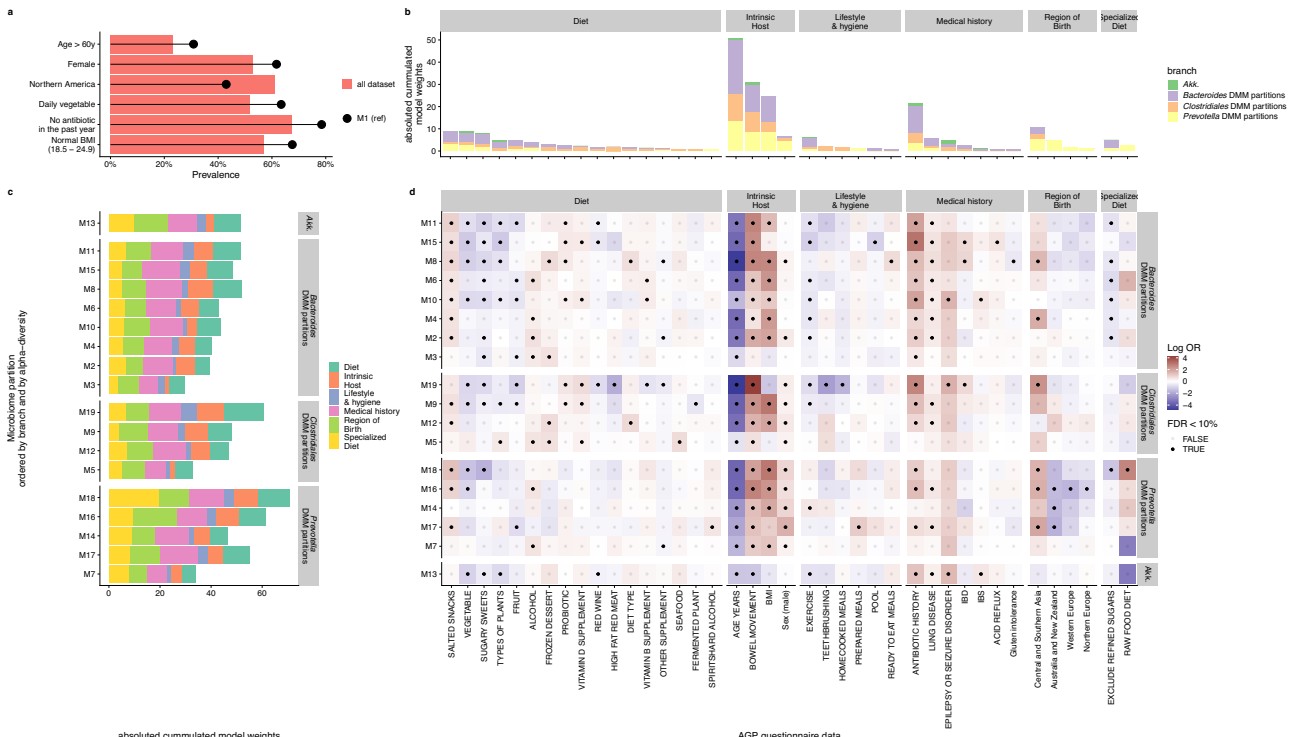

**Fig. 4 | Prediction of gut microbiome partitions from AGP metadata in relation to observed branches. a** Barplot showing the prevalence of a selection of AGP questionnaire answers in all AGP participants compared to AGP belonging to the gut microbiome partition M1. Partition M1, the highest alpha diversity and the structural root of branches, was used as the reference for multinomial logistic regression analysis. **b** Barplot showing predictors ranked by their cumulated model weight and depicted by categories. **c** Absolute model weights by predictor used in the model depicted among branches and partitions. Partitions were ranked based on gut microbiome increasing (top to bottom) alpha diversity (Shannon index) within each branch **d** Heatmap showing predictor log odd ratio (OR) as a function of partition grouped by branches. Significant OR were shown with a black dot (FDR < 10%). A positive log OR is associated with higher prevalence or intake. For antibiotic history positive log OR indicates that partition is associated with a shorter time since the last antibiotic intake.

environmental factors in our model. Furthermore, host-associated predictors model weight variations appeared to be specific to branch. For instance, along the *Prevotella* branch, from root to tip, we observed a model weight increase for BMI and a model weight decrease for age, while along the *Bacteroides* branch, we observed a model weight increase for bowel movements per day (Fig. 4d).

We also observed specific association with branches in a global manner that were not associated with partition position within branches. Specifically, sex (being male), region of birth (e.g., Australia & New Zealand, central and southern Asia) had a high coefficient weight for predicting partitions in the *Prevotella* branch, lung disease, and exercise frequency were most specific to the *Bacteroides* branch (Fig. 4d). Hence, the ability to predict a partition from AGP metadata was associated to overall observed branches structure.

Next, we sought to identify partitions that deviate from these overall predictors' gut microbiome differences (vs. partition M1) by considering the odds ratios of each predictor per partition (Fig. 4d). For instance, Clostridiales-enriched partition M5 exhibited a higher odd ratio in the number of types of plants (OR 1.59) and seafood frequency (OR 2.09) consumption. *Bacteroides*-enriched partition M15, M11, and Clostridiales-enriched partition M19 were mainly associated with diseases, and the *Akkermansia* partition was significantly associated with lower bowel frequency (Fig. 4d).

Taken together, our data showed that partitions exhibited both common and differential characteristics for both environmental and host factors that were branch-dependent or independent.

## Discussion

The high inter-subject variability of the human gut microbiome is well recognized and complicates both the study of association with factors

and response to treatment or dietary interventions. Multiple studies have partitioned human gut microbiomes with both supervised and unsupervised approaches but have suffered from low sample sizes or homogeneous cohorts. Here, we performed the largest-scale analysis to date (ca. 35,000 samples) to stratify the human gut microbiome across lifespan and populations. Using two complementary approaches (i.e., ordination and partitioning) on cross-sectional and longitudinal data, our analysis reveals that the human gut microbiome can be structured into partitions representing local ecological states that are dynamically linked within global gradients, referred to as microbiome branches.

Multiple studies have reported different microbiota states, using either supervised (alpha-diversity) or unsupervised (enterotypes) analysis. Using Dirichlet multinomial mixture models (DMM), ten states have been identified in a study of infants with developing gut microbiome from birth to four years[30], while (2-5) are typically detected in adults[28]. By contrast, Lathi et al. identified 32 gut microbiome partitions using specific, dominant features in adults[36] with the same approach we also used in our study. Then, while most studies analyze partitions independently, we further mapped them into branches, using a recently developed approach, PHATE, a non-linear dimensionality-reduction method that accounts for both local and global structures in a dataset[27].

Using a single (AGP) or multiple cohorts (CMD), we showed that similar numbers of partitions converge into two previously identified configurations (*Bacteroides* and *Prevotella* enriched) in adults and are centered around Clostridiales-enriched partitions, which were also enriched in *Bifidobacterium*, and characterized by a higher health index[35]. In adults, the analysis at the species level revealed that the central root was enriched in *F. prausnitzii, R. bromii, B. adolescentis*

while tips were enriched with respective members of *Bacteroides* or *Prevotella* and others such as *C. bolteae* which was found to be durably enriched following antibiotic intake[37,38]. A pseudotime analysis allowed us to further identify the decline in species along branches. For instance, *B. adolescentis* and *R. bromii* decreased earlier in the progression along the branches towards the tips than *F. prausnitzii*. Thus, pseudotime analysis allowed us to identify successive ecological states within branches.

The main difference between the two datasets was the detection in CMD of some partitions exclusively detected in infants, which were enriched in aerotolerant genera or *Bifidobacterium*, a common marker of the gut microbiome in early life, which is consistent with the under-representation of infants in the AGP dataset. Notably, multiple branches were detected in infants, which is consistent with a higher inter-subject gut microbiome variability observed in this population compared to adults[39]. Therefore, using complementary approaches, we identified various microbial states connected within branches, which differed in microbial composition, function, and potentially health status.

Previous studies identified a low-diversity *Bacteroides* microbiome state associated with obesity and systemic inflammation[18,19], which may be reflected in our study by the *Bacteroides*-enriched partition with the lowest diversity and associated with higher disease prevalence. This suggests that our analysis allowed us to capture both well-established and more recently identified partitions that could be of further interest. In addition, we observed that partitions with a similar alpha-diversity from different branches were associated with different factors, confirming the need to study the gut microbiome beyond alpha-diversity[40]. Extremities of both *Prevotella* and *Bacteroides* branches, which harbored the lowest gut microbiome diversity, differed in function compared to more central Clostridiales-enriched root in a respectively specific manner.

Next, we reasoned that partitions could represent gut microbiome states structured as a network with preferred paths, with individuals being more likely to transition between neighboring partitions than more distant ones. This hypothesis was supported by a network analysis performed without a priori knowledge (without coordinates of partitions) on a large dataset of subjects with time-series data (>700 subjects), which showed that indeed, partition switches occurred most frequently between neighboring partitions.

We confirmed previous findings of a low rate of switching between the *Bacteroides* and *Prevotella* branches[25,28,41]. Some partitions were more prone to shift to a specific partition over time, that were non-random inter-partition switches. This result suggests partitions may be connected through low-entropy ecological paths, which deserves further exploration with controlled time series studies. In addition, specific partitions appeared more stable than others, implying that some partitions may be stable ecological attractors while others would be transient states of the human gut microbiome. The least stable partitions were not associated with a lower diversity but instead with *Faecalibacterium:Bacteroides* ratio variation. However, it is unclear which factors contribute to these higher shift rates and whether the loss of *Faecalibacterium* precedes or results from the transition between partitions. A recent study identified *Faecalibacterium prausnitzii* among species associated with stability in healthy subjects[24]. In addition, we found that one of the lowest-diversity *Bacteroides* partition, which was more associated with diseases, antibiotics history, and higher bowel movement frequency, was both the most stable (i.e., little movement out of this partition) and was the most extreme (i.e., located at the tip) in its branch. Low diversity has been previously associated with instability[7,25]. Therefore, this could indicate that this partition represents a highly altered stable state, in which transition to another would require a significant force to shift to another state. Overall, we found intra-branch differential stability, which was not captured in previous studies with smaller sample sizes.

Subsequently, we reasoned that those partitions would be characterized by different host and environmental variables depending on branches. We confirmed results from previous studies identifying age, BMI, bowel movement frequency, and antibiotic intake as the top factors associated with gut microbiota variation[2–4]. Notably, differences between branches were observed, such as the *Prevotella*-enriched branch associated with the region of birth (a proxy for lifestyle and dietary habits) and the *Bacteroides*-enriched branch with medical history and lower exercise, and a lower bowel movement frequency was associated with the *Akkermansia* branch, in line with previous studies[42,43]. Specific associations were found within branches such as the association between the lowest diversity partition of the *Prevotella* branch with specialized diets. In line with our results, associations with diet were found to be partition-specific in a previous study[1], suggesting that insight into more refined structures allows for the identification of associations that may be obscured at the global level. Integrating partitions within a branch allowed us to better disentangle both microbiome-environment and microbiome-host associations.

Our study has several strengths, including the large size and highly heterogeneous population, which allowed us to uniquely partition highly variable gut microbiomes, and reproducing the findings from a heterogenous dataset to a homogeneous one. Furthermore, we showed the use of complementary and independent approaches to partitioning the gut microbiome by using both large cross-sectional ($n > 10,000$) and longitudinal data.

Our study has several limitations. First, health status, diet, and host parameters were either lacking or self-reported. Second, our study relied primarily on westernized adults and therefore needs to be complemented by underexplored populations. Third, the longitudinal study was not controlled and had varying stool collection periods on a limited number of subjects per partition, and should be reproduced on other longitudinal datasets. In addition, the American Gut project database is not representative of the global population, and the sub-cohort with multiple samples analyzed may be even less representative. Future more controlled longitudinal studies combining higher gut microbiome resolution and clinical data would allow further exploration of how partitions' composition and functional potential may vary with clinical outcomes.

Nevertheless, our study expands current knowledge of gut microbiota variation on an unprecedented sample size in various populations, health conditions, and ages. Potential future applications of our ecological framework include the study of the least explored transition from infancy to childhood, short-term challenges (diet, antibiotics), or longer-term ones (aging, disease) which will help identify the gut microbiome's transition to novel states that may guide microbiome-based approaches.

In conclusion, our study showed the relevance of structuring human gut microbiome data at a local and global level to better capture associations with health, diet, and lifestyle and identify transitions that may represent alternative states. Whether these ecological states and paths constrain the response of the human gut microbiome to changes in its environment warrants further investigation. This updated view of the human gut microbiome landscape provides a conceptual framework for moving towards precision nutrition and therapeutic approaches.

## Methods

### Studies datasets

**CuratedMetagenomicData (CMD).** We extracted taxonomic and functional tables from the curatedMetagenomicData R package[26] (version 3.0, release 2021), which aggregates read counts per species using MetaPhlAn3 and metabolic pathways per species using HUMaNN3 from 86 studies. Only stool samples counting more than five million reads and excluding one duplicated study (referred to as "LeChatelierE_2013") were retained. For taxonomic analyses, the

samples' read counts were sum-collapsed per genus and per species. The resulting feature table was rarefied to a depth of 1,000,000 counts per sample for alpha diversity analysis. Outliers analysis were performed following the criteria in ref. 44.

**American Gut Project (AGP).** In the AGP initiative, stool samples were collected at home and shipped at room temperature before microbial DNA extraction and 16 S rRNA amplicon sequencing, which were performed as previously described[4]. We used redbiom[45] to fetch data from Qiita[46]. 20,454 stool sample identifiers were available in the database on the date 2019 December 5th, within the Deblur-Illumina-16S-V4-100nt-fbc5b2 context. Analyses were performed as previously described[44]. In short, bioinformatic analysis was performed with QIIME 2019.10, bloom sequences were removed as previously described[47], and taxonomy was assigned using the GreenGenes database (v 13.5). We retained samples ≥1000 reads. 1579 samples were defined as technical outliers and excluded following the criteria in ref. 44. A genus count matrix with 16,021 samples was analyzed.

## Statistics and reproducibility
This study is a meta-analysis of multiple publicly available datasets, so the sample size was not predetermined using statistical methods. Instead, we try to include all available data and specified our criteria for selecting particular subsets of studies. The methodologies for data collection from individuals, such as whether the process was performed blind or not, can be found in the publications corresponding to each study. In addition, we used two large datasets where we applied same analyses to ensure findings reproducibility.

## Gut microbiome partitioning and branches
In the exploration cohort both studies' datasets, CMD and AGP, the 30 genera with the highest read mass were extracted for downstream analyses. Samples were partitioned using Dirichlet's multinomial mixture (DMM) modeling on the microbiota data[29]. We trained DMM models using five subsets from the whole dataset to reduce population bias. Each subset was sampled from the whole dataset with stratification by each combination of sex, geographic region of origin (region of birth for the AGP dataset), and age class (n up to 30 by strata). Subsampling was not limited to a single sample per subject. Of note, participants having at least two samples represented less than 5% of the participants. DMM modeling was performed for each subset constituting our training set, and the best model was picked up using BIC and Laplace minima, and majority vote (i.e., which optimal value of the k parameter was picked up more often). Partitions homogeneity were assessed using theta index extracted from DMM models. Low values of theta correspond to highly variable partitions. The whole dataset was modeled with DMM using the corresponding genera from the training dataset in the remaining dataset. Genera alpha weights for each DMM component from the CMD and AGP datasets were compared using hierarchical clustering on the Jensen-Shannon distance (Ward's method).

We assessed microbiome branch latent structure on the CMD and AGP datasets (genus relative abundance) using the PHATE algorithm (phateR version 1.0.7), with the gamma parameter set to zero[27] for visualization purposes and all other parameters set to default. Based on DMM genera alpha weight, we extracted microbiome branches.

For differential analysis at species level, abundances were centered log-ratio (CLR) transformed after adding a pseudocount of 1. Mann–Whitney tests were performed between the central partition and each partition corresponding to branch tips. For each comparaison, p values were adjusted for multiplicity by Benjamini-Hochberg false discovery rate correction.

## Longitudinal analysis
To assess stability patterns within partitions and switching between partitions over time, we extracted data from 745 participants of the AGP dataset who had provided at least two samples, resulting in 2998 samples. Partition stability was assessed by comparing the proportion of individuals who remained in their initial assigned partition and the proportion of individuals who remained in their randomly assigned partition. 100 random assignations were performed to compute a confidence interval for each partition. The percentile 95th for stability obtained from randomized assigned partitions was retained as a threshold for significatively.

Stability and proportion of change were computed for each partition. A network was built to show how individuals can change partitions as a function of time using igraph R package 1.3.5. Only edges representing more than 5% of the partition events (including switch and non-switch), to decrease visual noise, were shown. Fruchterman–Reingold algorithm was used to visualize the network.

## Health, demographic, and dietary associations with partitions and branches
To assess whether health, demographic, and diet participants' metadata were associated with gut microbiome partitions and branches in AGP, we fitted a multinomial log-linear model via neural networks using the nnet R package (version 7.3-15). Gut microbiome partitions were used as responses and metadata as predictors. The partition with the highest alpha-diversity assessed with the Shannon index was defined as the reference.

Categorical variables, like region of birth, were one-hot encoded. Binary predictors, like disease self-declaration, were coded by 0 to 1 (e.g., "Diagnosed by a medical professional" encoded to 1, "I do not have this conditions" encoded to 0, other categories were considered as missing values). Continuous predictors, like Age and BMI, were scaled from 0 to 1. Ordered predictors, like food groups frequency, were coded from 0 (never) to 1 (daily). Antibiotic history was considered as an ordered factor from "I have not taken antibiotics in the past year" to antibiotic intake within the "week" of fecal sample collection. In short, 100 health, demographic and dietary predictors were used to build the model. Missing values were replaced by the average computed for each predictor.

Log odds ratios by predictor and gut microbiome partition and their respective p-values were extracted from the resulting model using broom_helpers R package (version 1.2.1). False discovery rate was computed by predictors. All statistical analyses were computed using R software version 3.6[48].

## Functional analysis of gut microbiome branches
The OXYTOL database v 1.3 was used to associate each microbial genus to an aerotolerant or obligate anaerobic metabolism[49]. A score resulting on dividing aerotolerant aggregated read count by anaerotolerant aggregated read count was performed for each sample.

Differentially-abundant metabolic pathways were identified using multinomial regressions via neural networks in songbird version 1.0.3[50]. This analysis was performed on the curatedMetagenomicData pathways, after collapsing (i.e., summing per sample) the read counts across taxa, hence resulting in a list of coefficients for each pathway, indicating their strength of association with each model's target. Models of pathway enrichment were run on sample subsets belonging the DMM partitions associated to branch tips and DMM partition identified as central that serve as reference group. The performance of neural network models was monitored using tensorboard (version 1.15.0) in terms of cross-validation error and log-likelihood loss during training/testing iteration for a 70/30 ratio split performed on the full sample set. Optimal model solutions were reached for trials involving run parameters varying for number of epochs and batch sizes. Those models and parameters retained for post-processing analyses of the coefficients are available in configurations files (supplementary material and code) used by prep_songbird (version 1.0), which allowed generating the subsets and songbird command lines in QIIME2. For

each model, the coefficients were ranked to get the top 10 pathways found most strongly associated with each model target. These pathways were used as numerators to calculate log ratios for the samples composing the targeted and reference partitions. 21 "nucleotide biosynthesis" pathways were used as a denominator because they represented ubiquitous pathways in microbial metabolism, which were present in 98% of the samples.

## Pseudotime analysis

To confirm the ordering of our partitions along enterobranches, we performed a pseudotime analysis using Wishbone, which orders samples along a two-way trajectory branching off from a common root[34]. Although Wishbone is suited to avoid short-circuit between trajectories for complex datasets, we performed the analysis on a subset of the samples consisting of the central partition m8 (expected trunk) and the partitions enriched in *Prevotella* (m4, m12, m13, m14, m16, m17) and *Bacteroides* (m2, m3, m5, m20, m21) (expected branches). The trajectory was detected using a randomly selected sample from the central partition (m8) as a starting point (EGAR00001421178_9002000001557655LL) as well as the first two components resulting from the eigendecomposition operated on the diffusion matrix (see Wishbone methods). Results are represented on a t-SNE map computed after a projection of the data on the three first components of a PCA (perplexity=30; seed=666).

## Data availability

All data used in this study are publically available. CuratedMetagenomicData are available at https://github.com/waldronlab/curatedMetagenomicData. The AGP data used in this study are available in Qiita under the study ID 10317, and the associated sequences can be found under EBI accession ERP012803. Source data for all main figures are provided as supported data files. Source data are provided with this paper.

## Code availability

Source codes used in this study are available from GitHub (http://github.com/tapj/branches) and Zenodo (https://doi.org/10.5281/zenodo.7801644).

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

## Acknowledgements
The authors wish to thank Heleen de Weerd, Martin Balvers, and Jolanda Lambert for bioinformatics support. This work was funded by Danone Nutricia Research.

## Author contributions
The authors' responsibilities were as follows: J.T., P.V., M.D. conceived the study. J.T., D.M., R.K., S.J.S., P.V., M.D. designed the analysis; J.T., F.L.: analyzed the data. A.C.; J.T., P.V., M.P., M.D.: interpreted the results; J.T., F.L., M.D., and P.V.: wrote the first version of the manuscript; and all authors: read, contributed, and approved the final manuscript.

## Competing interests
A.C., M.P. are employees of Danone Nutricia Research. J.T., M.D., P.V. were employees of Danone Nutricia Research when this project was conducted. F.L., D.M., and S.-J.S are supported through a collaborative research agreement with Danone Nutricia Research. The other author reports no competing interests.
