## [Peer Review File · Nature Communications]

REVIEWER COMMENTS

Reviewer #1 (Remarks to the Author):

The manuscript by Julien Tap et al. presents a meta-analysis aiming at exploring latent structures of the human gut microbiome across the human lifespan. This is accomplished by considering a total of 35,000 samples coming from two main public available resources: the CuratedMetagenomicData package (17,000 samples associated with shotgun metagenomes) and the American Gut Project (AGP; 16,000 samples associated with 16S rRNA data). A specific analysis is also conducted by considering longitudinal data from 745 individuals. Authors identify three major gut microbiome branches, with each branch that can be further partitioned in the adult population.

Overall, I found the manuscript interesting for the microbiome community. However, I have few comments:

1. From my understanding the manuscript relies completely on public available data and existing methodologies for data analysis. I think authors should better state novelties of the present paper with respect to existing literature.
2. I think the taxonomic analysis on shotgun data should be extended in some way to species-level resolution.
3. It seems that the longitudinal analysis is conducted on 16S data only. I think an extension to shotgun data would be important. Authors may use some of the longitudinal data that are available in the CuratedMetagenomicData package that is already analyzed in this paper.
4. I cannot access the GitHub page with the source code (<https://github.com/tapj/branches>).

Reviewer #2 (Remarks to the Author):

This paper globally analyzes a large amount of data from the CMD database (around 18K) and also 16K from the American gut project (AGP) in order to map the latent landscape of GBM states and analyze transitions of individuals longitudinally across this space. Overall, I think this is a powerful analysis approach that combines the strengths of both continuum analysis and global visualization (with PHATE) and discretized analysis and characterization with DMMs.

I have a few comments intended to increase the clarity of the work.

1. While the authors use a manifold learning approach, namely PHATE for visualizing the data. they don't use the related PHATE.cluster which performs spectral clustering (first proposed in Ng. et al 2003) on the diffusion operator computed for PHATE which would be a natural way to discretize this data. Alternatively the newer multi scale PHATE [Kuchroo et al. Nat Biotech 2022] also performs discretization along with the visualization . I think this approach should be tested since it operates on the same latent (diffusion operator) representation of the data.

One of the issues is seen in Figure 1 part H where PHATE is showing around 9 branches which seem partitioned into a plethora of clusters, without great motivation. One could perform pseudo time analysis (as in Moon et al. 2019) on the branches instead and provide a continuum placement on the branch.

2. The authors don't provide a complete explanation of the use of DMM which is not common in biology compared to community detection for example. What is the rationale for using it? I am not saying there isn't but it should just be stated for a self-contained manuscript.

3. Figure 2 shows results from the AGP dataset which look significantly different from the CMD dataset. I surmise the differences are due to 16s sequencing and the inclusion of only adult samples in AGP which simplifies the overall structure. I think a further examination of these differences would be interesting and of value to the readership.

For instance overall, why would adults have simpler GBMs?

4. 16s seems to give species counts while metagenomic sequences would be a collection of genes at each level. It would be informative to mention how distances in these approaches would be related to one another. For instance could one perform some kind of species count inference (like CIBERSORTX) to make it closer to 16s analysis?

5. It would be interesting to draw longitudinal trajectories on the PHATE maps to see the extent of switching in Figure 3 without relying on the partitioning granularity. Are longitudinal trajectories coherent along branches, do they have a tendency to go from branch bases to branch tips?

6. For predictive analysis it would be interesting to perform mutual information analysis (as in Moon et al 2019) to see which species or genes vary continuously with branches, and whether these can be extrapolated for prediction.

Minor:

The term ordination is somewhat confusing. Please define it or use an alternative term.

REVIEWER COMMENTS

Reviewer #1 (Remarks to the Author):

The manuscript by Julien Tap et al. presents a meta-analysis aiming at exploring latent structures of the human gut microbiome across the human lifespan. This is accomplished by considering a total of 35,000 samples coming from two main public available resources: the CuratedMetagenomicData package (17,000 samples associated with shotgun metagenomes) and the American Gut Project (AGP; 16,000 samples associated with 16S rRNA data). A specific analysis is also conducted by considering longitudinal data from 745 individuals. Authors identify three major gut microbiome branches, with each branch that can be further partitioned in the adult population.

Overall, I found the manuscript interesting for the microbiome community. However, I have few comments:

1. From my understanding the manuscript relies completely on public available data and existing methodologies for data analysis. I think authors should better state novelties of the present paper with respect to existing literature.

Thanks for your comment. **We have strengthened the part of novelty in the discussion.**

Lines 476-480: “Our study has several strengths, including the large size and highly heterogeneous population, which allowed us to uniquely partition highly variable gut microbiomes, and reproducing the findings from a heterogeneous dataset to a homogeneous one. Furthermore, we showed the use of complementary and independent approaches to partition the gut microbiome by using both cross-sectional and longitudinal data”

2. I think the taxonomic analysis on shotgun data should be extended in some way to species-level resolution.

Thanks for your comment. **We did differential analysis to compare branch tips versus central partitions for species on adults from CMD at species level.**

We added some sentences about differential analysis at species level in the results section and discussion and added a supporting table. (Table S3)

Results

Lines 168-179:

“A differential analysis at species level (CLR transformed) on adult subjects revealed multiple differences between the central root (M8) and tips of branches (**Supplementary Table S3**). For instance, the partition closest to the tip in the *Bacteroides* branch (m21) was enriched in *Bacteroides stercoris* and *Clostridium boltea* while depleted in *Bifidobacterium adolescentis* and *Ruminococcus bromii*. Meanwhile, the partition closest to the tip of *Prevotella* branch (m16) was enriched in *Prevotella coprii*, *Prevotella stercorea* while depleted in *R. bromii* and *Eubacterium hallii*. The partition encompassing the *Bifidobacterium* branch (m24) was enriched in *Bifidobacterium bifidum*, *Bifidobacterium breve* and depleted in *R. bromii* and *Dorea longicatena*. The ‘aerobic and aerotolerant’

branch was enriched in *Enterococcus faecalis*, *Streptococcus epidermis*, *Streptococcus mitis* and depleted in *R. bromii*, *B. adolescentis*, *F. prausnitzii*. Overall, the core was most often enriched in *B. adolescentis* and Clostridiales (*R. bromii*, *F. prausnitzii*) compared to tips”.

Discussion

Lines 415-422:

“In adults, the analysis at the species level revealed that the central root was enriched in *F. prausnitzii*, *R. bromii*, *B. adolescentis* while tips were enriched with respective members of *Bacteroides* or *Prevotella* and others such as *C. boltea* which was found to be durably enriched following antibiotic intake (FitzGerald et al 2022, Palleja et al 2018). A pseudotime analysis allowed us to further identify the decline in species along branches. For instance, *B. adolescentis* and *R. bromii* decreased earlier in the progression along the branches towards the tips than *F. prausnitzii*. Thus, pseudotime analysis allowed us to identify successive ecological states within branches”

3. It seems that the longitudinal analysis is conducted on 16S data only. I think an extension to shotgun data would be important. Authors may use some of the longitudinal data that are available in the CuratedMetagenomicData package that is already analyzed in this paper.

Thanks for your comment. We agree that assessing / confirming the observation of transition between partitions on other datasets than AGP is relevant. The strength of AGP is the number of subjects (n~800). The major limitations of most longitudinal studies are 1) the low number of subjects and 2) therefore the subsequent heterogeneity between studies when pooled. The additional issue of CMD is that the order of visit is not known in most of the studies, hampering the analysis. See reply below for Reviewer 2.

From CuratedMetagenomicData, A total of four studies (Hall AB, 2017; HMP 2019 IBD; HMP 2019 T2D; Yassour M, 2016) reported order of visit, consisting of 132 subjects, which is not sufficient for the analysis as we partition gut microbiome.

We strengthened this limitation in the discussion

Line 484-492

“Our study has several limitations. First, health status, diet, and host parameters were either lacking or self-reported. Second, our study relied primarily on westernized adults and therefore needs to be complemented by underexplored populations. Third, the longitudinal study was not controlled and had varying stool collection periods on a limited number of subjects per partition, and should be reproduced on other longitudinal datasets. In addition, the American Gut project database is not representative of the global population, and the sub-cohort with multiple samples analyzed may be even less representative. Future more controlled longitudinal studies combining higher gut microbiome resolution and clinical data would allow further exploration of how partitions' composition and functional potential may vary with clinical outcomes”.

4. I cannot access the GitHub page with the source code

Thanks for your comment. We have now verified that the code is publicly accessible.

<https://github.com/tapj/branches>

Reviewer #2 (Remarks to the Author):

This paper globally analyzes a large amount of data from the CMD database (around 18K) and also 16K from the American gut project (AGP) in order to map the latent landscape of GBM states and analyze transitions of individuals longitudinally across this space. Overall, I think this is a powerful analysis approach that combines the strengths of both continuum analysis and global visualization (with PHATE) and discretized analysis and characterization with DMMs.

I have a few comments intended to increase the clarity of the work. 1. While the authors use a manifold learning approach, namely PHATE for visualizing the data. they don't use the related PHATE.cluster which performs spectral clustering (first proposed in Ng. et al 2003) on the diffusion operator computed for PHATE which would be a natural way to discretize this data. Alternatively the newer multi scale PHATE [Kuchroo et al. Nat Biotech 2022] also performs discretization along with the visualization. I think this approach should be tested since it operates on the same latent (diffusion operator) representation of the data.

One of the issues is seen in Figure 1 part H where PHATE is showing around 9 branches which seem partitioned into a plethora of clusters, without great motivation. One could perform pseudo time analysis (as in Moon et al. 2019) on the branches instead and provide a continuum placement on the branch.

Thanks for your suggestions. We first applied Phate 2 on CMD. We clustered samples at coarse level and detected 14 clusters that could be mapped on Phate coordinates (see figure below).

We performed global comparison between DMM-based clusters and Phate-multiscale by computing a confusion matrix, which we visualized by a hierarchical clustering heatmap (Ward).

Overall, we found a good correspondence between PHATE multiscale cluster and DMM partitions, notably for partition and clusters located close to branches tips. For example, DMM#16 correspond nicely to PHATE cluster #808 and DMM #21 with PHATE cluster #1371. However, DMM #19 has been separated into seven PHATE clusters while DMM #2, #3, #1, #6, #8 have been grouped into PHATE cluster #0. However, we see that cluster #0 seems to be distributed into several branches which correspond to different DMM partitions.

We performed also the analysis on the AGP dataset to check whether PHATE-multiscale found the same pattern on cluster #0 on AGP. We found that cluster #0 1) aggregates several DMM and 2) visually belongs to different branches. This observation needs to be further explored by performing another round of phate multiscale clustering of samples assigned to cluster #0. However benchmarking DMM partitions and phate algorithm is not the focus of this paper. Therefore, we did not include that analysis in the paper.

One of the issues is seen in Figure 1 part H where PHATE is showing around 9 branches which seem partitioned into a plethora of clusters, without great motivation.

The fact that we obtain 9 branches in CMD is related to the presence of infants in CMD. In adults, the major branches are *Prevotella* and *Bacteroides* centered around Clostridiales, while infants are split into more but less distinct branches, which is consistent with previous studies showing that infants have a higher inter-subject variability in gut microbiome versus adults. This has been reviewed by Derrien et al, 2019 (DOI:<https://doi.org/10.1016/j.tim.2019.08.001>)

One could perform pseudo time analysis (as in Moon et al. 2019) on the branches instead and provide a continuum placement on the branch.

Following the reviewer #2 suggestion, we performed a pseudotime analysis that is meant to order the data points along trajectories, that in our case consist of the characterized branches. The authors used Wanderlust, a tool that orders cell samples using seriation along every detected trajectory to represent their developmental continuum. The authors also refer to additional pseudotime-ordering tools, such as PAGA (Wolf et al., 2019) or Wishbone (Setty et al., 2016). All should be able to confirm the ordering of our partitions along branches. For the practical reason of applying pseudotime-ordering on our dataset, we chose the latter tool, Wishbone. No attempt was made with any other tool.

Moreover, Wishbone was notably developed to detect more efficiently branches between inferred trajectories, which is likely to affect our dataset. Indeed, as pointed out by the reviewer there are ca. 9 branches and several clusters that we aim to discretize in pseudotime-ordering, and the below t-SNE maps indicate numerous possible such short-circuit between partitions (see Figure S7).

Wishbone computes diffusion maps that consist of eigenvectors/eigenvalues obtained from a decomposition performed on a matrix of affinities between samples. This matrix results from the normalization of the distances to the neighbors of each point measured using unsupervised (10-)nearest neighbors search on our dataset reduced to using the (general) loadings of a (3) PCA components. Wishbone employs t-SNE for plotting and thus, we shall first represent on the t-SNE plot the partitions of each major PHATE branch, before attempting to compute the diffusion maps (see Figure S7).

We integrated a pseudo time analysis in main text with two supplemental figures (Fig S7 and Fig S8)

Lines 180-198: “We further tested whether branching composition continuums could be retrieved using an alternative and a complementary, pseudo-time method called Wishbone (Setty et al 2016), to provide an ordering of samples along the continuous branches. Wishbone is tailored to detect a trajectory based on a t-SNE ordination, that bifurcates from a common root into two branches, and along which the samples are ordered. Hence, we ran Wishbone on a t-SNE ordination computed from 8,356 samples encompassing the main branches (*Bacteroides* and *Prevotella*) as well as the central partition (m8). Consistent with the PHATE analysis, Wishbone detected the central m8 partition as the root from which the two *Bacteroides* and *Prevotella* branches deviated (**Supplementary Fig.6**). At genus level, from root to the tips we observed that *Bifidobacterium* and *Ruminococcus* declined along the branches, followed by *Faecalibacterium* while *Bacteroides* and *Prevotella* increased in their respective branch, confirming our observations from the DMM partitions (**Supplementary Fig. 7a**). Reporting the abundance dynamics at the species level along the Wishbone trajectory revealed a succession for different species of *Bacteroides* (**Supplementary Fig. 7b**) including *B. ovatus* and *B. fragilis* only appearing at the very end of the Wishbone trajectory. Interestingly, *F. prausnitzii* and *B. bifidum* exhibited a gradual and sharp decrease in abundance along the trajectory, respectively. The

partitions were ordered in a similar way to PHATE along these branches, keeping in notably the partitions located respectively on the tips of the two branches. (Supplementary Fig. 7c). Overall, we showed that DMM-based local partitions can be ordered within the global structure, called branches, revealed by PHATE and confirmed by Wishbone.”

2. The authors don't provide a complete explanation of the use of DMM which is not common in biology compared to community detection for example. What is the rationale for using it? I am not saying there isn't but it should just be stated for a self-contained manuscript.

Thanks for your comment. Dirichlet Multinomial Mixtures (DMM) models is a commonly used clustering approach in gut microbiome (see for instance Beller et al 2021, Costea et al 2018, Ding & Schloss 2014, Holmes et al 2012, Roswall et al 2021, Stewart et al 2018, Vieira-Silva et al 2020). DMM models assume that microbiota data originate from multinomial sampling, and they use a Bayesian framework with Dirichlet prior distributions to estimate K components (or clusters) (Holmes et al., 2012). This explicit sampling scheme can model sparse data. In addition, data being directly modelled as proportions, DMM models are thus also adapted for compositional data. DMM models output a probability to belong to each component for each sample, which allows to assign it to the most probable component, but also provides a confidence indicator upon its classification. Another advantage of this type of model is that it allows to build components with different dispersions, which is much more realistic than algorithms assuming equal dispersions between clusters (e.g. implicit assumption for K-means). In addition, the model can easily be used to classify a new subject that was not included in the original modelling, which was useful in our case since we performed the modelling on a sub-sampling of the data to balance age categories and alleviate the overrepresentation of North Americans and Europeans in CMD.

We have now added a sentence in the first section of Results

Lines 99-102: “In short, DMM models are adapted for compositional data, provide a confidence indicator upon samples’ classification, allow for building partitions with different dispersions, and can be used to classify a new sample that was not included in the original modelling”.

3. Figure 2 shows results from the AGP dataset which look significantly different from the CMD dataset. I surmise the differences are due to 16s sequencing and the inclusion of only adult samples in AGP which simplifies the overall structure. I think a further examination of these differences would be interesting and of value to the readership. For instance overall, why would adults have simpler GBMs?

Thanks for your comment.

We checked whether partitions were database-specific. We found that Clustering partitions did not result in dataset specific clustering (Figure 2a. Figure 2 shows that AGP and CMD are similar (Ward clustering and Jensen-Shannon distance between DMM-based partitions alpha weights from both datasets. There was no significant difference between CMD and AGP datasets (PERMANOVA, $p > 0.05$). However, AGP lacks the infant-associated branches / partitions, consistent with low number of infants in AGP.

To better evaluate the differences between both datasets and given the computational variability between 16S and Shotgun to profile species level, we compared both datasets based on genus level.

The major differences are the lack of Akkermansia branch in CMD and lack of infant branch in AGP (explained by the lack of infants)

In adults, which is highly complex and the major branches are Prevotella and Bacteroides centered around Clostridiales, while infants are most diverse branches, which is consistent with previous studies showing that infants have a higher inter-subject variability in gut microbiome versus adults. This has been reviewed by Derrien et al, 2019 (DOI:<https://doi.org/10.1016/j.tim.2019.08.001>)

In short, many studies have highlighted a lower bacterial diversity, a lower functional complexity, and a higher degree of interpersonal variation in gut bacterial diversity between infants than between adults (Bäckhed F. et al. Cell Host Microbe. 2015; 17: 690-703; Laursen M.F. et al. mSphere. 2016; 1e00069-15; Yatsunenko T. et al. Nature. 2012; 486: 222-227; Chu D.M. et al. Nat. Med. 2017; 23: 314-326). Below is an example of Bäckhed et al study, in which beta-diversity (diversity amongst subjects) is higher in infants.

ACTION TAKEN: We added a sentence in the discussion

Lines 426-428: “Notably, multiple branches were detected in infants, which is consistent with a higher inter-subject gut microbiome variability observed in this population compared to adults (Bäckhed et al 2015).”

4. 16s seems to give species counts while metagenomic sequences would be a collection of genes at each level. It would be informative to mention how distances in these approaches would be related to one another. For instance could one perform some kind of species count inference (like CIBERSORTX) to make it closer to 16s analysis?

Thanks for your comment. Please find below some information about species-level assignment

We did not actually analyze shotgun metagenomics data at gene level but already aggregated at different taxonomic level from MetaPhlAn outputs. So we compared directly taxonomic composition between 16S data derived from QIIME ASVs and shotgun data computed by MetaPhlAn available from CuratedMetagenomics database. Thank for the suggestion about computing distances between 16S and gene levels from metagenomics data, this methodological comparison is out of scope for this study. Our goal was to replicate global structure from the two datasets that derived from different method but not to compare method directly.

5. It would be interesting to draw longitudinal trajectories on the PHATE maps to see the extent

of switching in Figure 3 without relying on the partitioning granularity. Are longitudinal trajectories coherent along branches, do they have a tendency to go from branch bases to branch tips?

Thanks for your comment. We understand from your comment that the way of the extent of switching between, partitions was not sufficiently explained in the paper. In short, the network on longitudinal data was done without taking into account of the order of partitions within Phate branches. It was done by unsupervised approach. The fact that switching (network based) occurs between partitions that kept order between partitions reinforce that partitions are ecological states. We now stress more how network data support the fact that partitions are ecological states in abstract, results and discussion

Abstract

Line 32: “An unsupervised network analysis from longitudinal data from 745 individuals showed that partitions exhibited connected gut microbiome states rather than over-partitioning”

Results

Lines 289-290: “Of note, this network was performed in an unsupervised fashion, (i.e not integrating the partition coordinates depicted in Fig. 2d).”

Discussion

Lines 443-446: “This hypothesis was supported by a network analysis performed without a priori knowledge (without coordinates of partitions) on a large dataset of subjects with time-series data (>700 subjects), which showed that indeed, partition switches occurred most frequently between neighboring partitions.”

We also performed additional analysis. We selected subjects with the highest number of samples in AGP. One individual has a time-scale of 5 years, early in which they underwent surgery. We observed that this individual follows a pattern from the tip of Bacteroides enriched branch to the root following the procedure.

One individual has variation on 1 month. We observed a quite high stability as during one month this individual had a gut microbiome composition close to the Bacteroides enriched branch tip.

The last individual has variation of 3 years and more exactly two time series separated by one year. This individual was quite stable in Bacteroides enriched branch while one year later (with no samples within) this individual was found to be in Prevotella enriched branch with an instable pattern.

Overall, over several time scale, individuals' gut microbiome varied along branches. We observed that microbiome in subjects with subjects with largest time series moved along branches (Fig Supp). From one subject with surgery (top panel), post-surgery variation of gut microbiome indicates that this individual moved from the tip to the root of Bacteroides branch. For the second subject (2nd panel), gut microbiome remained in Bacteroides branch, or for the third subject (third panel) changed branch within 1 year (without known exposure to treatment or intervention).

We did not add this analysis in the manuscript as we think that 3 subjects are not representative enough to include in this study.

6. For predictive analysis it would be interesting to perform mutual information analysis (as in Moon

et al 2019) to see which species or genes vary continuously with branches, and whether these can be extrapolated for prediction.

Thanks for your comment. As for reviewer 1, we did differential analysis to compare branch tips versus central partitions for species on adults from CMD at species level. We added a supporting table. (Table S3)

ACTION TAKEN: We added some sentences about differential analysis at species level in the results section and discussion

Results

Lines 168-179:

“A differential analysis at species level (CLR transformed) on adult subjects revealed multiple differences between the central root (M8) and tips of branches (**Supplementary Table S3**). For instance, the partition closest to the tip in the *Bacteroides* branch (m21) was enriched in *Bacteroides stercoris* and *Clostridium bolteae* while depleted in *Bifidobacterium adolescentis* and *Ruminococcus bromii*. Meanwhile, the partition closest to the tip of *Prevotella* branch (m16) was enriched in *Prevotella coprii*, *Prevotella stercorea* while depleted in *R. bromii* and *Eubacterium hallii*. The partition encompassing the *Bifidobacterium* branch (m24) was enriched in *Bifidobacterium bifidum*, *Bifidobacterium breve* and depleted in *R. bromii* and *Dorea longicatena*. The ‘aerobic and aerotolerant’ branch was enriched in *Enterococcus faecalis*, *Streptococcus epidermis*, *Streptococcus mitis* and depleted in *R. bromii*, *B. adolescentis*, *F.prausnitzii*. Overall, the core was most often enriched in *B. adolescentis* and Clostridiales (*R. bromii*, *F. prausnitzii*) compared to tips”.

Discussion

Lines 415-422:

“In adults, the analysis at the species level revealed that the central root was enriched in *F. prausnitzii*, *R. bromii*, *B. adolescentis* while tips were enriched with respective members of *Bacteroides* or *Prevotella* and others such as *C. bolteae* which was found to be durably enriched following antibiotic intake (FitzGerald et al 2022, Palleja et al 2018). A pseudotime analysis allowed us to further identify the decline in species along branches. For instance, *B. adolescentis* and *R. bromii* decreased earlier in the progression along the branches towards the tips than *F. prausnitzii*. Thus, pseudotime analysis allowed us to identify successive ecological states within branches”

We also performed functional analysis. This was already included in the Results section “Low-diversity tips of branches display potential functional shifts “ lines 206-229. In short, extremities of both *Prevotella* and *Bacteroides* branches, which harbored the lowest gut microbiome diversity, differed in function compared to more central Clostridiales-enriched root in a respectively specific manner.

Minor:

The term ordination is somewhat confusing. Please define it or use an alternative term.

Thanks for your comment. We have added a sentence in introduction lines 54-56

“Ordination methods, techniques such as principal coordinates analysis that aim at representing or analyzing multivariate data in fewer dimensions (Hui et al 2015), have been used to visualize and describe microbiome data.”

REVIEWERS' COMMENTS

Reviewer #3 (Remarks to the Author):

Disclaimer: I was asked to assess whether the responses to the comments of reviewer #2 are satisfactory, as reviewer #2 is not able to re-review themselves.

To this end, and in brief, I do think that the authors have provided thoughtful and clear responses to all the original comments of reviewer #2.